# Hole Quality Observation in Single-Shot Drilling of CFRP/Al7075-T6 Composite Metal Stacks Using Customized Twist Drill Design

Jebaratnam Joy Mathavan [1,2], Muhammad Hafiz Hassan [1,*], Jinyang Xu [3,*] and Gérald Franz [4]

1   School of Mechanical Engineering, Universiti Sains Malaysia, Nibong Tebal 14300, Malaysia
2   Department of Engineering Technology, Faculty of Technology, University of Jaffna, Kilinochchi Premises, Ariviyal Nagar, Kilinochchi 44000, Sri Lanka
3   State Key Laboratory of Mechanical System and Vibration, School of Mechanical Engineering, Shanghai Jiao Tong University, Shanghai 200240, China
4   Laboratoire des Technologies Innovantes, UR UPJV 3899, Avenue des Facultés, Le Bailly, 80025 Amiens, France
*   Correspondence: mhafizhassan@usm.my (M.H.H.); xujinyang@sjtu.edu.cn (J.X.)

**Abstract:** In the modern aircraft manufacturing industry, the use of fiber metal stack-up material plays an important role. During assembly, these stack-up materials need to be drilled, and single-shot drilling is the best option to avoid misalignments. This paper discusses hole quality in terms of hole edge defects and hole integrity with respect to tool geometry. In this study, tungsten carbide (WC) twist-type drills with various geometric features were fabricated, tested, and evaluated. Twenty custom twist drill bits with primary clearance angles ranging from 6° to 8°, chisel edge angles from 30° to 45°, and point angles from 130° to 140° were fabricated. The CFRP and Al 7075-T6 were stacked up, and a feed rate of 0.05 mm/rev and spindle speed of 2600 rev/min were used for all drilling experiments. The experimental array was constructed using response surface methodology (RSM) to design the experiments. The impact of factors and their importance on hole quality were investigated using analysis of variance (ANOVA). The study demonstrates that the primary clearance angle, followed by the chisel edge angle, is the most important factor determining hole quality. As a function of tool geometry, correlation models between exit delamination and burr height were developed. The findings suggested that, within the range of parameters examined, the proposed correlation models might be utilized to predict performance measures. For drilling CFRP/AL7075-T6 stack material in a single shot, the ideal twist drill geometry was determined to be a 45° chisel edge angle, 8° primary clearance angle, and 130° point angle. For optimum drill geometry, the discrepancy between the expected and actual experiment values was 0.11% for exit delamination and 9.72% for burr height. The findings of this research elucidate the relationship between tool geometry and hole quality in single-shot drilling of composite-metal stacks, and more specifically, they may serve as a useful, practical guide for single-shot drilling of CFRP/Al7075-T6 stack for the manufacture of aircraft.

**Keywords:** single-shot drilling; CFRP/Al stacks; hole quality; optimization; twist drill; ANOVA

## 1. Introduction

Composite materials have gained prominence during the past few years as a means of reducing the weight of aircraft structures. In actuality, 52% of the Airbus A350's total structural materials and 57% of the Boeing 787's major structure are composites [1], and carbon fiber-reinforced polymer (CFRP) is the most widely utilized fiber [2]. Fibrous composites and metallic alloys, such as titanium/aluminum alloys, are widely employed in stack form in the current aerospace sector to gain enhanced mechanical properties and function for components requiring energy-saving features [3–5]. Composites have

enormous application potential in a variety of modern commercial aircraft, including the Airbus A350 and Boeing 787 Dreamliner [6,7]. Although there are many fibers and metals, the superior qualities of CFRP, Al, and Ti are attractive for this application and typically, while manufacturing a stack up, a composite panel is placed on top of the metal part [8]. To produce different geometric characteristics for improved product integrity, reliability, life cycle, and secure assembly with other components, hole drilling has long been a standard procedure in the manufacturing sector [9,10]. When it comes to the machining of lightweight metals and composites, this method has been particularly important in the automotive and aerospace industries [11,12]. Given that metals are isotropic and fibers are anisotropic, single-shot stack drilling involving these two components is extremely difficult and can result in a variety of tool and hole damage. This research intends to determine appropriate tool geometry improvements in regard to hole quality.

Delamination is a main issue related to the drilling of fiber-reinforced composite materials, which tends to decrease the structural integrity of the material [13]. Delamination damage is a force-associated failure that can be classified into two types [14], namely peel-up delamination and push-out delamination [1]. Peel-up delamination is caused by tool geometry [15], whereas push-out delamination is caused by thrust force inserted by the drill point [16]. Because of the irreparable nature of delamination damage, the composite laminate has to be rejected when delamination reaches a certain extent [17]. As a consequence of the inhomogeneity of fibrous composites, measurement of the extent of delamination becomes challenging. The hardness of carbon fibers induces abrasive wear at the cutting edge of the drill, which in turn increases the thrust force during drilling, finally leading to delamination [18]. On the other hand, high drilling thrust force can also be generated by increased drill feed [19]. Conversely, research carried out on dry drilling of CFRP/Al/CFRP by ref. [20] found that an increase in feed rate resulted in a positive influence on entrance delamination. Faster chip evacuation is caused by the selection of higher axial feed, thus reducing contact time and friction [21]. Delamination initiates from the CFRP matrix laminate side of the stack-up material. Although a defect-less CFRP element is formed to a near net shape, delamination at the exit of the hole is inevitable during the drilling of rivet holes. Delamination weakens the structural consistency of the CFRP part in terms of tensile and bearing strength [22], and it may also reduce the fatigue life [23].

Burr formation is a challenging factor in the aircraft industry related to multi-material stack drilling because rough edges (commonly named burrs) on fastener holes can induce stress concentrations, which may initiate corrosion, fatigue failure, reduction in the life of the aircraft [24], injuries to workers, and can reduce the functionality of the components [25]. Although burr height is the commonly measured parameter for evaluating burrs, burr thickness causes more deburring costs than burr height [26]. Usually, exit burr height and exit burr root thickness are noticeably larger than those of entrance burrs. This is primarily because the burr formed at the entrance is caused by a tearing action, which includes a bending process followed by lateral extrusion or clean shearing, whereas the exit burr is formed by plastic deformation of the workpiece material in front of the chisel edge, without the material being cut [24]. This is because the ductility of the aluminum alloy increases due to thermal softening from the higher machining temperature at higher spindle speeds. The increase in ductility allows the material to flow easily and at this stage, as the tool exits from the hole, the aluminum material is stretched and pushed out to form a burr along the edge of the hole [27]. An increasing point angle and larger helix angle tend to reduce burr root thickness and burr height [28].

Because of the tight tolerance of the hole diameter in the assembly process of the aircraft, the difference in hole diameters between the stack-up materials during drilling is an important problem. This difference in diameter occurs because of the different properties of the stack-up materials, including their elasticity modulus, which leads to different elastic deformations that make it difficult to control the difference in hole diameters between the stack-up materials [29–32]. Even if the hole in one material of the stack is undersized

(hole diameter < tool diameter) or oversized (hole diameter > tool diameter), a reparation technique needs to be applied that usually adds extra costs and time to the assembly procedure. Soo et al. [31] used two types of twist drill designs, namely flat point and double cone, to drill CFRP/Al stack material in a single shot, revealing that the double cone drill bit design helped control the difference in hole diameter between both stack materials due to thin chip formation and easy evacuation of chips through the drill flute during the drilling process. Benezech et al. [33] mentioned the importance of the axial rake angle and the included angle on drilled hole quality during CFRP-Al stack drilling. These authors kept a constant axial rake angle throughout the length of the cutting edge, since it is advantageous for good quality drilling. The combination of 135° point angle and 30° rake angle gave the optimum twist drill geometry to attain high quality holes in stack-up material. Kuo et al. [34] investigated the number of margins in the twist drill bit design that significantly influenced the diameter of the hole, regardless of the feed rate and drilling technique. Triple-margin drills gave less vibration and larger contact with the machined surface, yielding smaller hole diameter variation. The higher ductility of the aluminum alloy resulted in a change to long, twisted helical chips, but they were tightly folded as the drill progressed into the stack. Accordingly, oversized holes can simply arise when the drilling operation is executed in dry conditions.

To date, the influence of the geometric parameters of twist drills on hole quality have not been reported by previous researchers. This research work highlights the single effects of the geometric parameters of the drill bit on hole quality. This paper is written in a format that provides a brief introduction to the relationship between drilling parameters and hole quality in Section 1. Section 2 describes the materials and methods used in this work. Section 3 contains the results and discussion, and finally, Section 4 delivers the conclusion of this research work.

## 2. Materials and Methods

### 2.1. Worpiece Materials

CFRP and 7075-T6 aluminum alloy (Al7075-T6) were the stack materials used in this study. The CFRP composite specimen had a total laminate thickness of 3.25 mm and was made up of 26 unidirectional plies, each 0.125 mm in thickness. Hexcel Composite Company's carbon/epoxy prepreg was used to make the 26 unidirectional plies with a stacking sequence of $[45/135/90_2/0/90/0/90/0/135/45_2/135]_s$. The CFRP laminate was then covered with a 0.08 mm thin layer of glass/epoxy woven fiber at the top and bottom to prevent delamination at both the entrance and exit of the hole during drilling. As a result, the final thickness of the entire composite panel, including the paint application, was 3.587 mm. The CFRP was compressed using a vacuum pump and controlled atmospheric conditions during the curing process. The autoclave was equipped with a prepared mold to keep the laminate. The temperature was raised to 180 °C during the curing cycle at a rate of 3 °C/min and maintained for 120 min. The temperature was then gradually brought back to normal temperature. The entire cycle was carried out in an autoclave at a pressure of 700 kPa and the laminate was packed in a vacuum bag that was depressurized to 70 kPa. Because of the curing recipe's application, the nominal fiber volume was 60%. The mechanical and physical characteristics of the stack materials employed in this work are compiled in Table 1.

**Table 1.** Mechanical properties of CFRP and Al7075-T6.

| Properties | Tensile Strength [MPa] | Elasticity Module [GPa] | Elongation [%] | Flexural Strength [MPa] | Density [g/cm³] | Thickness [mm] |
|---|---|---|---|---|---|---|
| CFRP | 2723 | 164 | 1.62 | 1500 | 1.601 | 3.587 |
| Al7075-T6 | 558 | 71.7 | 13 | - | 2.597 | 3.317 |

### 2.2. Cutting Tool Fabrication

The drill bit type was a combination of drill and countersink. The drill's diameter was 4.826 mm and the diameter of the countersink was 10 mm. Due to its excellent resistance to wear while drilling abrasive materials like CFRP, a sintered rod of tungsten carbide (WC) was chosen as the drill bit material. The tungsten carbide rod was made up of 93.36 wt% WC and 6.64 wt% Cobalt (Co). Since tungsten carbide has a Vickers hardness value of 1625 HV and density of 14.35 g/cm$^3$, both of which are much higher than those of the workpiece, it was selected as the drill bit material. Helitronic Tool Studio version 1.9.216.0 software (Walter Maschinenbau GmbH, Garbsen, Germany) was used to design the drills with special custom drill geometry. A particular wheel must perform numerous consecutive operations while grinding with a cutting tool. These operations include pointing, gashing, and clearing. In the program, a chisel edge angle of 30° to 45° was set for the gashing process and a primary clearance angle of 6° to 8° was selected for the clearance phase. The point angle was finally established from 130° to 140° during the pointing phase. Figure 1a–c demonstrates the manufacturing procedure and the wheel type used to modify the twist drill design using a CNC grinding machine (Walter Maschinenbau GmbH, Garbsen, Germany).

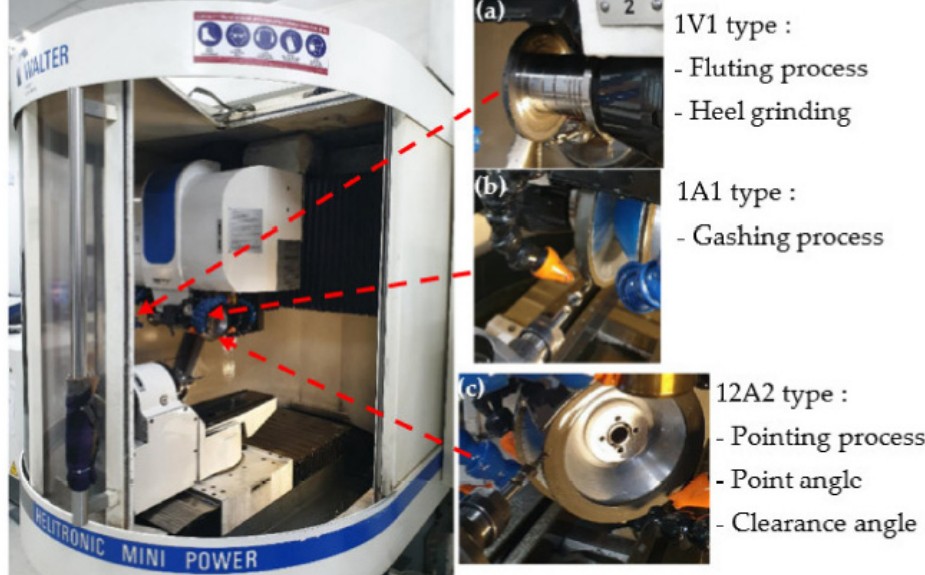

**Figure 1.** Location of grinding wheel for the tool fabrication: (**a**) fluting wheel, (**b**) gashing wheel, (**c**) clearance/point angle wheel.

### 2.3. Drilling Process

Using a computer numerical control (CNC) machine (Fanuc Robodrill T21iFLb), which has a variable spindle speed up to 10,000 rev/min and spindle drive motor of 3.7 kW at a continuous rating, the drilling of the stack material was carried out. For a regular rate, the feed rate can range from 1 to 30 mm/min, and for a high transverse rate, the feed rate can range from 48 m/min (*x*, *y*, and *z* axes). Drilling was performed in a single shot, starting at the CFRP panel and moving to the Al7075-T6 panel. During drilling, the stack panels were slotted into the fixture and clamped. To evaluate the major impact of the customized twist drill geometry, a feed rate of 0.05 mm/rev and spindle speed of 2600 rev/min were chosen for all runs in this study. In this experiment, dry drilling conditions were employed to simulate the drilling process that actually occurs during panel manufacturing. The angles of the standard twist drills, with variations in the three aforementioned angles, are summarized in Table 2. Design of experiment (DOE) was used to design the experimental process. DOE is a popular technique for constructing the number of experiments needed

to establish the statistical validity of the relationship between the input and output of independent variables. Twenty trials were administered in accordance with ref. [35].

**Table 2.** Experimental factors at different levels of chisel edge angle (A), primary clearance angle (B), and point angle (C).

| Level | Chisel Edge Angle [°] | Primary Clearance Angle [°] | Point Angle [°] | Spindle Speed [rev/min] | Feed Rate, [mm/rev] |
|---|---|---|---|---|---|
| Minimum | 30 | 6 | 130 | | |
| Midpoint | 37.5 | 7 | 135 | 2600 | 0.05 |
| Maximum | 45 | 8 | 140 | | |

### 2.4. Hole Edge Defect Measurement

The quality of a drilled hole in the aircraft industry can be defined based on the hole edge defects and hole integrity. Delamination in the CFRP phase and burr formation on the aluminum phase are the major hole edge defects. Poor hole edges can contribute to stress formation and rivet joint damage during mounting.

### 2.4.1. Exit Delamination

Delamination in this research was evaluated at the exit side of the CFRP laminate. Figure 2a shows the sample CFRP panel condition at the exit side after the drilling process. The laminate was assessed using an Alicona InfiniteFocus optical microscope at $20\times$ magnification to observe the delamination at the exit side of the CFRP panel in detail (Figure 2b). To measure the value of the delamination of the CFRP laminate at the end of the drilling process, a delamination factor was introduced. To make sure that the delamination was within the specification limits according to OEM standards, the delamination in the bore of the drilled hole at the exit hole face of the CFRP section must be less than 2 mm per side for laminates with a thickness of less than 5 mm. The images from the Alicona InfiniteFocus optical microscope were investigated using ImageJ software in order to determine the area of nominal value and the damaged area. The delamination factor, $F_{d-exit}$, was calculated based on the ratio of the damaged area to the nominal area, as shown in Equation (1).

$$F_{d-exit} = \frac{A_{max}}{A_{nom}} \tag{1}$$

where $A_{nom}$ is the nominal area of the drilled hole and $A_{max}$ is the damaged area of the composite laminate after the drilling process.

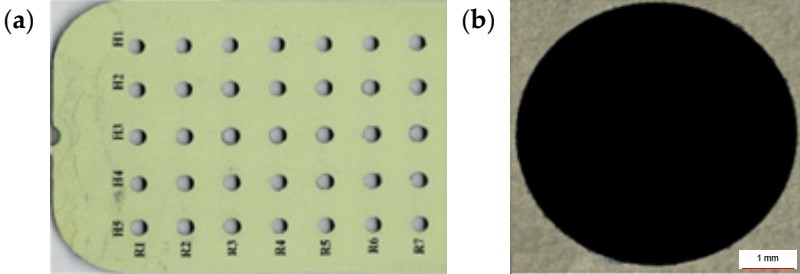

**Figure 2.** Observation of (**a**) exit (**b**) close-up view of delamination at the exit using an Alicona InfinateFocus optical microscope.

### 2.4.2. Burr Height

Minimizing burr formation is a vital criterion in drilling. The current study focused only on exit side burrs, since these generally lead to further processes such as dismantling, deburring, and reassembly of the stack, while entry side burrs are not significant because of the compaction force from the CFRP laminate. The primary factors that affect burr

formation are cutting parameters, tool geometry, and workpiece materials. The smallest burrs at the hole edge were observed by ref. [36] when the drilling feed rate was increased. The tendency for burr formation may increase if the material has moderate ductility, since the material tends to elongate because of the produced heat during the machining process.

Exit burr formation in this study was evaluated using an Alicona Infinite Focus optical microscope with a magnification of 20×, as shown in Figure 3. The optical measurement system was a non-contact type that accomplished the task without creating any surface damage. The detailed maximum burr formation measurement is shown in Figure 4. The maximum burr height was identified from the 3D diagram obtained from the Alicona InfiniteFocus optical microscope, and the highest burr point is marked by a red line, as shown in Figure 4c.

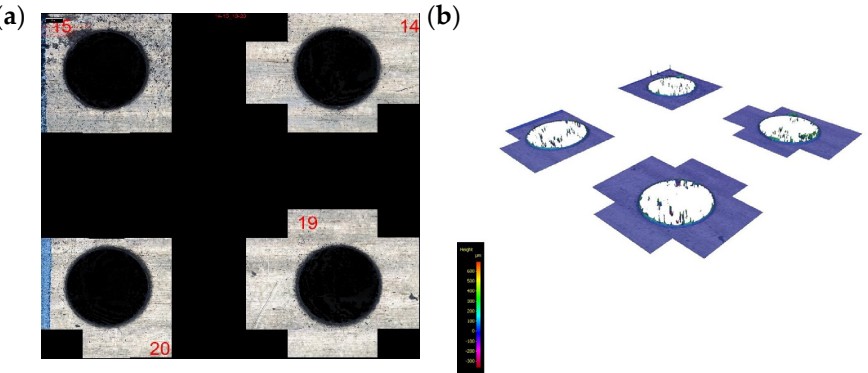

**Figure 3.** Type of burr formation observed under the Alicona Infinite Focus optical microscope: (**a**) uniform burr formation, (**b**) rolled-back burr formation.

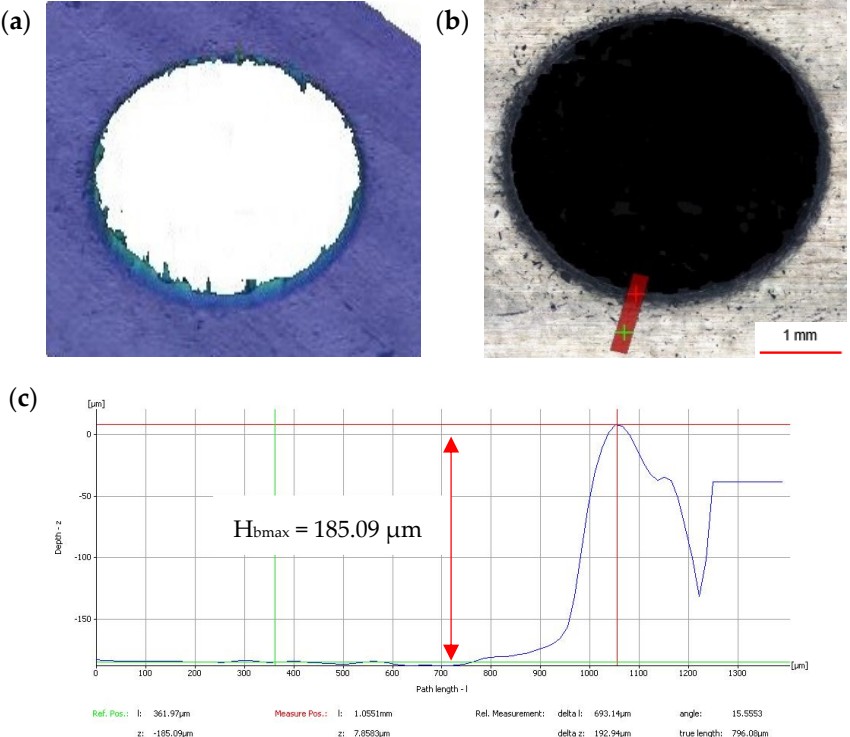

**Figure 4.** Measurement process of maximum burr formation: (**a**) 3D observation of burr formation, (**b**) maximum burr formation identification, and (**c**) maximum burr formation measurement.

*2.5. Hole Integrity*

Hole integrity is defined based on hole circularity or hole roundness and the difference in hole diameters between materials. Hole circularity defects and large differences in hole diameters between the stacking materials will also interrupt the assembly process, which in succession will increase the quantity of scrap panels. These parameters are very important and must be scrutinized frequently according to the requirements of the customer.

2.5.1. Hole Diameter Error

Ensuring the minimum difference in diameter between holes in CFRP and Al, which have different material properties, is one of the main tasks while drilling a stack-up material. The definition of hole diameter error is the difference of measured hole diameter to the nominal diameter. In the case of the stack-up material in this research work, hole diameter error was defined as the difference between the measured holes of CFRP and Al (Al7075-T6). A Crysta-Plus M443 coordinate measuring machine (CMM) with a probe size of 2 mm and accuracy in measuring error of $3.0 + 4 L/1000$ µm was used in this research work to measure the hole diameter errors for both CFRP and Al7075-T6. It can function in the *x*-axis, *y*-axis, and *z*-axis, commonly termed the three orthogonal axes, in a three-dimensional coordinate system. After the probe touched the surface, the point positions acted as the input and the data were transmitted to MCOSMOS v3 software.

Then, the software employed the *x*-axis, *y*-axis, and *z*-axis coordinates of every discrete point to find the mean diameter of the hole. The workpiece sample was raised up by a block, clamped, and placed in a position where it could be reached for measurements of the *x*-axis, *y*-axis, and *z*-axis. The coordinate system was arranged in a way to choose a datum point as a reference. The measurement of hole diameters started with the CFRP panel. For laminates ranging in thickness from 3 to 10 mm, the probe position should be in the center of the laminate, according to OEM standards. The probe in this research work was moved towards the center of the drilled hole and then downwards into half of the hole depth of the laminate thickness, as shown in Figure 5a. The positions of 0°, 90°, 180°, and 270° points were obtained as four reference points during the measurement to confirm the consistency of this procedure, as shown in Figure 5b. A circle appeared on the screen and the diameter was recorded. These steps were repeated for Al7075-T6 until the diameters of all the holes of the sample were measured, as shown in Figure 6. The following Equations (2)–(4) were used to calculate the hole diameter error for each panel and also between the laminates.

$$\varepsilon_{cfrp} = d_m - d_{nom} \tag{2}$$

$$\varepsilon_{al\ 7075} = d_m - d_{nom} \tag{3}$$

$$\varepsilon_{cfrp/al\ 7075} = d_{cfrp} - d_{al\ 7075} \tag{4}$$

where $\varepsilon_{cfrp}$ is the error for CFRP panel; $\varepsilon_{al\ 7075}$ is the error for Al7075-T6 panel; $\varepsilon_{cfrp/al\ 7075}$ is the difference in diameter between stack laminates; $d_m$ is the measured diameter; $d_{nom}$ is the nominal diameter; $d_{cfrp}$ is the measured diameter for CFRP; and $d_{al\ 7075}$ is the measured diameter for Al7075-T6 panel.

2.5.2. Hole Circularity

Hole circularity was measured using the Crysta-Plus M443 CMM in the same way as the hole diameter was measured. The only difference between the measurement of hole diameter and that of hole circularity was the number of points obtained for consideration. At least 40 points should be measured to obtain the least square diameter and circularity of a hole at a given depth [37]. A sample measurement and the point distribution to obtain hole circularity are shown in Figure 7. The hole circularity values of CFRP and Al7075-T6 were individually obtained from the information given in Figure 6 after all 40 points were touched.

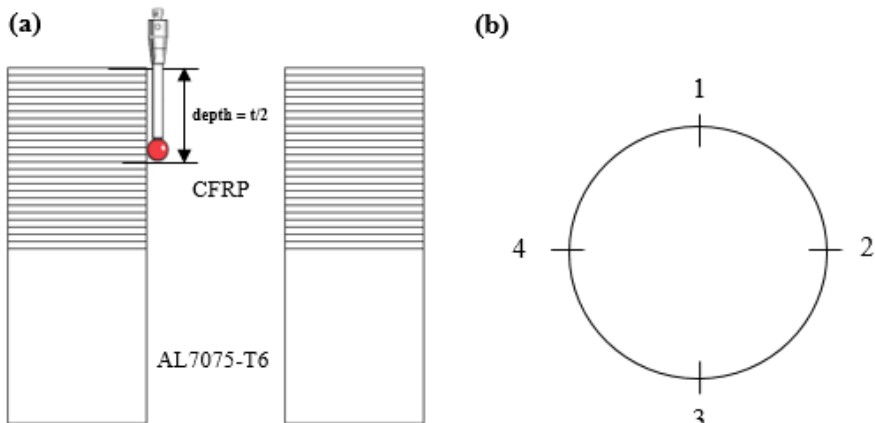

**Figure 5.** (**a**) Position of probe during measurement of each laminate CFRP and Al7075-T6 (**b**) point of contact for hole diameter measurement and hole circularity measurement.

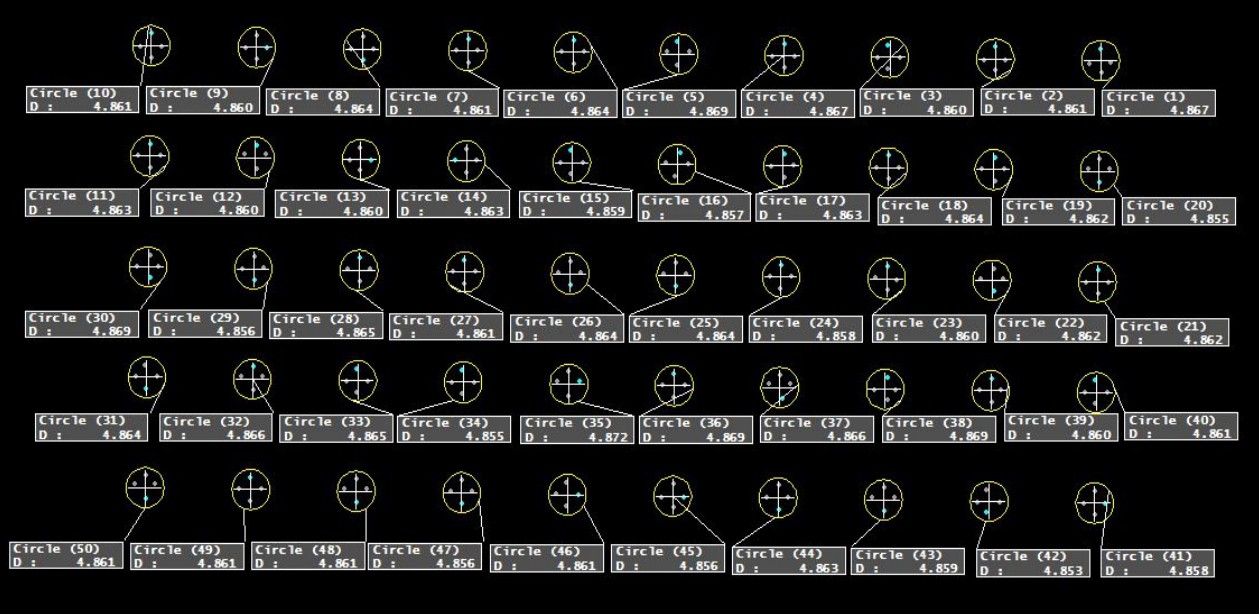

**Figure 6.** Example of hole diameter measurement for Al7075−T6 panel.

*2.6. Response Surface Methodology (RSM)*

RSM [35] has several advantages over Taguchi's method and is a crucial tool for optimizing a product or process and solving resilient design problems [38]. RSM was used to modify the twist drill geometry for single-shot drilling of the stack-up material. In this optimization study, interactions between the input variables were evaluated. The most common of all second-order designs, the central composite design (CCD), was used in the RSM design. The CCD comprises a full factorial design (2k) with 2k of axial or star points and center points, where k is the number of factors [39]. Between −1 and +1, there are three levels of variables. Table 3 lists the parameters that were selected for this investigation along with their coding levels. The formula CCD = $2^k$ + 2k + 6, was used to produce the number of experiment runs, where k was the number of components with replications at the design center. A quadratic model was applied to the optimization to fit and estimate the minimal point. The mathematical model for each answer was created using these data points, as illustrated in Equation (5) [40,41].

$$Y = \beta_0 + \sum_{i}^{k} \beta_i X_i + \sum_{i=1}^{k} \beta_{ii} X_i^2 + \sum_{i=1}^{k} \sum_{j=1}^{k} \beta_{ij} X_i X_j \tag{5}$$

where, $Y$ is the predicted response; $X_i$ and $X_j$ are the input variables; $\beta_0$ is an offset term; $\beta_i$, $\beta_{ii}$, and $\beta_{ij}$ are the interaction coefficients of linear, quadratic, and second-order terms.

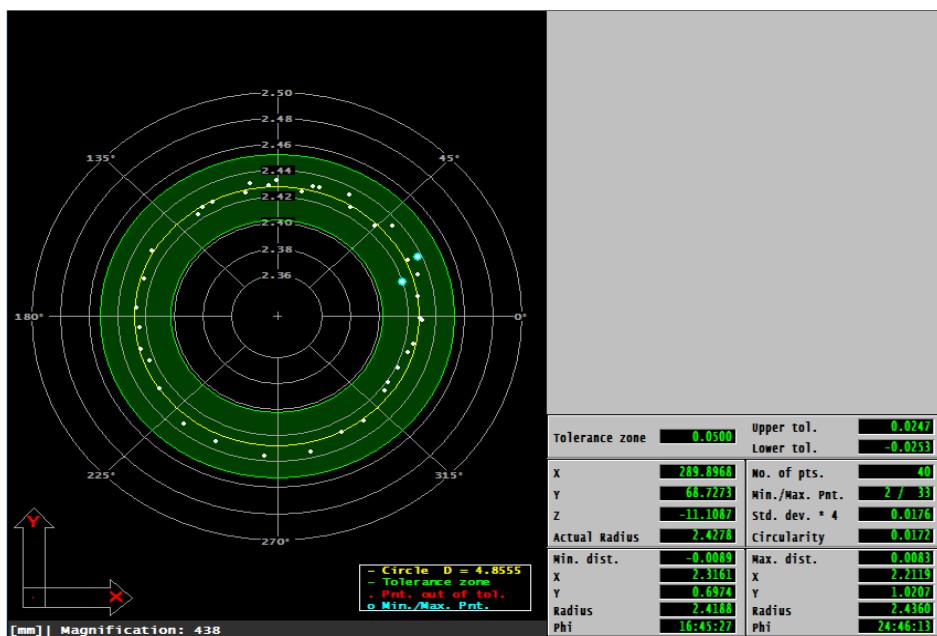

**Figure 7.** Example of hole circularity measurement of CFRP panel.

**Table 3.** The parameters selected for RSM investigation along with their coding levels.

| Input Variables | Lower Level (−1) | Coded Level (0) | Higher Level (+1) |
|---|---|---|---|
| Chisel Edge Angle [A°] | 30 | 37.5 | 45 |
| Primary Clearance Angle [B°] | 6 | 7 | 8 |
| Point Angle [C°] | 130 | 135 | 140 |

Design Expert 14 software was used to choose the regression models for the results based on the highest-order polynomials, significant additional terms, and lack of aliased models. The regression models were created in terms of coded and real components, with the best fitting of the quadratic equation or other transformation for all relevant model variables taken into consideration at values of *p*-values less than 0.05 [42]. Through the perturbation plot, the responsiveness of independent variables was determined. To determine the relationships between parameters, two significant components were chosen to create the 3D response surface. The intended aim, whether to minimize or enhance the output, was determined in accordance with the level's range, following the optimization process. The software then produced each factor's optimal value and reaction. The outcome of the experiment was then compared to the outcome of the regression models, which were constructed using the best value possible for each factor.

## 3. Results and Discussion

### 3.1. Exit Delamination Analysis

Figure 8a,b shows the minimum and maximum delamination, respectively, found at the CFRP panel's exit hole. Although the point angle of the drill bit was increased from 130° to 140°, the delamination for the entire run was within the permitted limit. This amply demonstrates that, during any of the tests, there was never a clear indication of delamination at the CFRP exit hole when the point angle was increased within the range. In a similar manner, Senthil Kumar et al. [43] used 118° and 130° of point angle drills to examine the effects of point angle on tool performance when drilling composite/Ti stack. It was determined that the higher point angle (130°) drills outperformed those with the

lower point angle (118°), based on tool wear and chip evacuation analysis. Geng et al. [1] demonstrated that the drilling thrust force exceeds the critical thrust force when exit delamination occurs. Since the composite panel's critical thrust force was not reached in this research, the entire run was conducted below it.

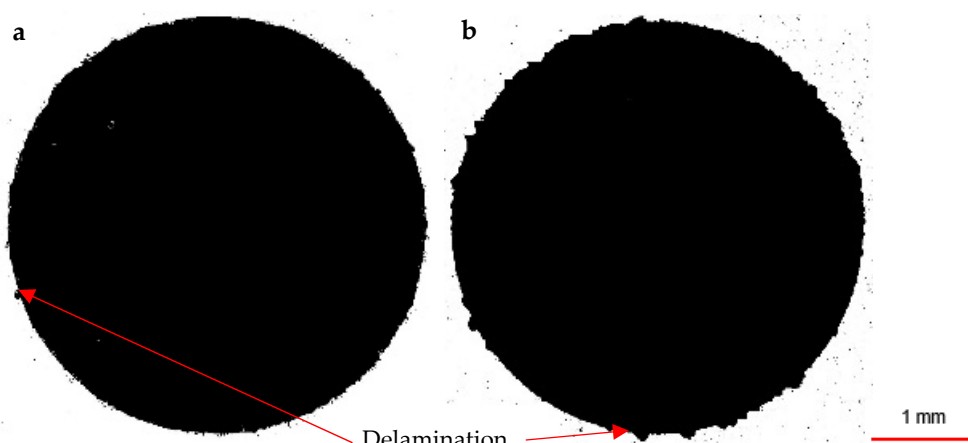

**Figure 8.** (**a**) Minimum and (**b**) Maximum delamination at the exit hole of the CFRP panel for all runs.

Figure 9 shows the detailed exit delamination factor $F_{d\text{-exit}}$ values for the entire run. The values acquired for each trial appeared to be nearly identical on the graph. The typical $F_{d\text{-exit}}$ value ranged from 1.0038 to 1.0196. Since every value was below the tolerance level advised by aerospace manufacturers in accordance with OEM standards, it is evident that the range of the drill geometry in this study had no impact on the $F_{d\text{-exit}}$ value. However, ref. [44] reported that twist drills are less efficient than core drills since the thrust force is not much focused on the middle of the drill bit and cutting edges, but usually distributed over the periphery of the bit.

### 3.1.1. Regression Model and ANOVA

To obtain the lowest residuals between the anticipated and actual values for delamination, the regression model for the response was enhanced using a quadratic model. The final empirical model for the actual causes of delamination at the CFRP panel's exit hole ($Y_1$) is shown in Equation (6).

$$Y_1 = 3.37924 - 2.34867e^{-3}A - 0.41360B - 0.011186C + 1.53654e^{-3}BC + 0.014163B^2 \quad (6)$$

For ($Y_1$), the *F*-value in the ANOVA analysis was 8.66 and the probability value (*p*-value) was less than 0.05, as shown in Table 4. Furthermore, the *p*-value of 0.9214 indicated that the lack of fit was related to pure error and was not significant. The model's significant value and the lack of fit's non-significant value supported the validity of the log-transformed model. The point angle was insignificant, despite the fact that the percentage of contribution (PC) for each model term, A, B, BC, and $B^2$, had a considerable impact on exit delamination, with values of 43.7%, 10.5%, 8.4%, and 20.4%, respectively. The values of the $R^2$, adjusted $R^2$ (Adj $R^2$), and predicted $R^2$ (Pred $R^2$) coefficients were used to assess the model's goodness of fit. The Adj $R^2$ value of 0.7053 and Pred $R^2$ value of 0.6885 were in reasonable accord as the discrepancy was less than 0.2.

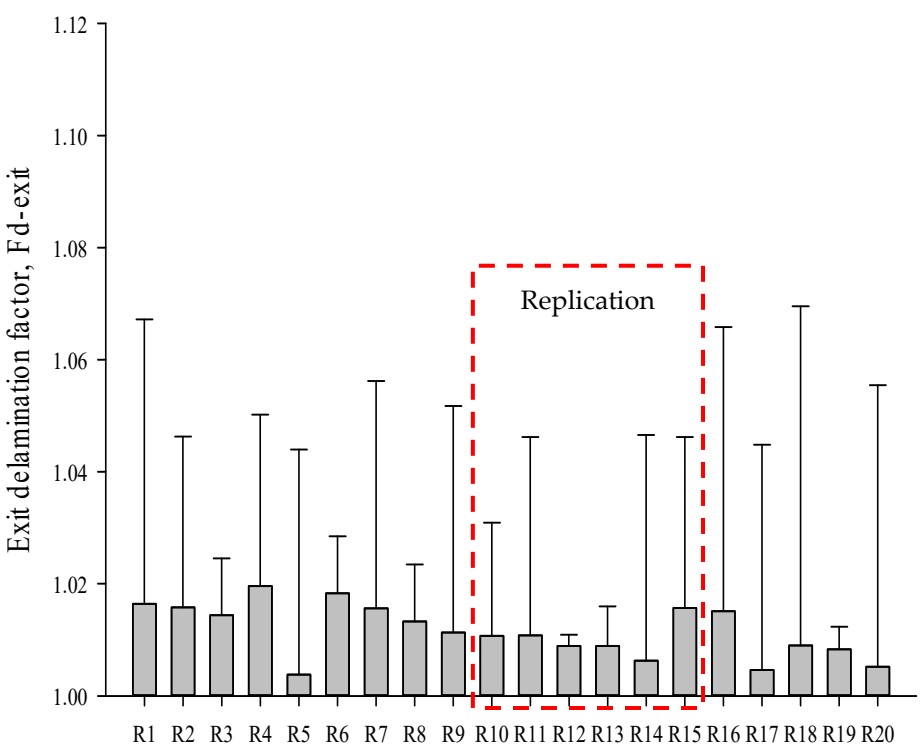

**Figure 9.** Exit delamination factor values of CFRP for all runs.

| | R1 | R2 | R3 | R4 | R5 | R6 | R7 | R8 | R9 | R10 |
|---|---|---|---|---|---|---|---|---|---|---|
| $F_{d\text{-exit}}$ | 1.0164 | 1.0158 | 1.0144 | 1.0196 | 1.0038 | 1.0183 | 1.0156 | 1.0133 | 1.0113 | 1.0107 |
| | **R11** | **R12** | **R13** | **R14** | **R15** | **R16** | **R17** | **R18** | **R19** | **R20** |
| $F_{d\text{-exit}}$ | 1.0108 | 1.0089 | 1.0089 | 1.0063 | 1.0157 | 1.0151 | 1.0046 | 1.0090 | 1.0083 | 1.0052 |

**Table 4.** Pooled ANOVA of model for exit delamination of CFRP panel.

| Source | Sum of Squares | df | Mean Square | F Value | *p*-Value | PC (%) | |
|---|---|---|---|---|---|---|---|
| **Model ($Y_1$)** | 0.0032506 | 5 | 0.0004911 | 8.66 | 0.0015 | | Significant |
| Chisel edge angle (A) | 0.001695 | 1 | 0.001695 | 29.87 | 0.0002 | 43.7% | |
| Primary clearance angle (B) | 0.0004053 | 1 | 0.0004053 | 7.15 | 0.0217 | 10.5% | |
| Point angle (C) | $3.393 \times 10^{-5}$ | 1 | $3.393 \times 10^{-5}$ | 0.6 | 0.4556 | 0.9% | |
| BC | 0.0003246 | 1 | 0.0003246 | 5.72 | 0.0357 | 8.4% | |
| $B^2$ | 0.0007918 | 1 | 0.0007918 | 13.96 | 0.0033 | 20.4% | |
| Residual | 0.000624 | 11 | $5.673 \times 10^{-5}$ | | | 16.1% | |
| Lack of Fit | 0.0001586 | 6 | $2.643 \times 10^{-5}$ | 0.28 | 0.9214 | | not significant |
| Pure Error | 0.0004654 | 5 | $9.308 \times 10^{-5}$ | | | | |
| Cor Total | 0.0038746 | 16 | | | | | |
| Std. Dev. | $7.53 \times 10^{-3}$ | | $R^2$ | | 0.7974 | | |
| Mean | 1.04 | | Adj $R^2$ | | 0.7053 | | |
| C.V. % | 0.73 | | Pred $R^2$ | | 0.6885 | | |
| PRESS | $9.59 \times 10^{-4}$ | | Adeq Precision | | 10.446 | | |

A signal-to-noise ratio greater than 4 is preferred when measuring signal-to-noise with adequate precision [45]. Since a strong signal was indicated by the ratio (Adeq Precision) of 10.446, this model was utilized to navigate the design space. According to ref. [46], $R^2$ should be at least 0.80 for a model to fit the data well. The correlation coefficient ($R^2$) and adjusted coefficient (Adj. $R^2$) values in this instance were 0.7974 and 0.7053,

respectively, demonstrating the significance of the fit of the RSM model and its potential for response prediction.

When the actual value gained through experimentation was compared with the predictions of the model, as shown in Figure 10a, it can be observed that the points were evenly split by a 45-degree line, which proved the fit of the model. This established the reliability of the regression model modifications for predicting exit delamination. The graph demonstrates that, when compared to the expected value predicted by the empirical model, the response of the experimental data was mostly contained within the range of allowable deviations, as shown in Figure 10b. When drilling stack-up material, the CFRP delamination could be estimated using the regression model created here. The standard error estimation (SEE) result was 0.0060795.

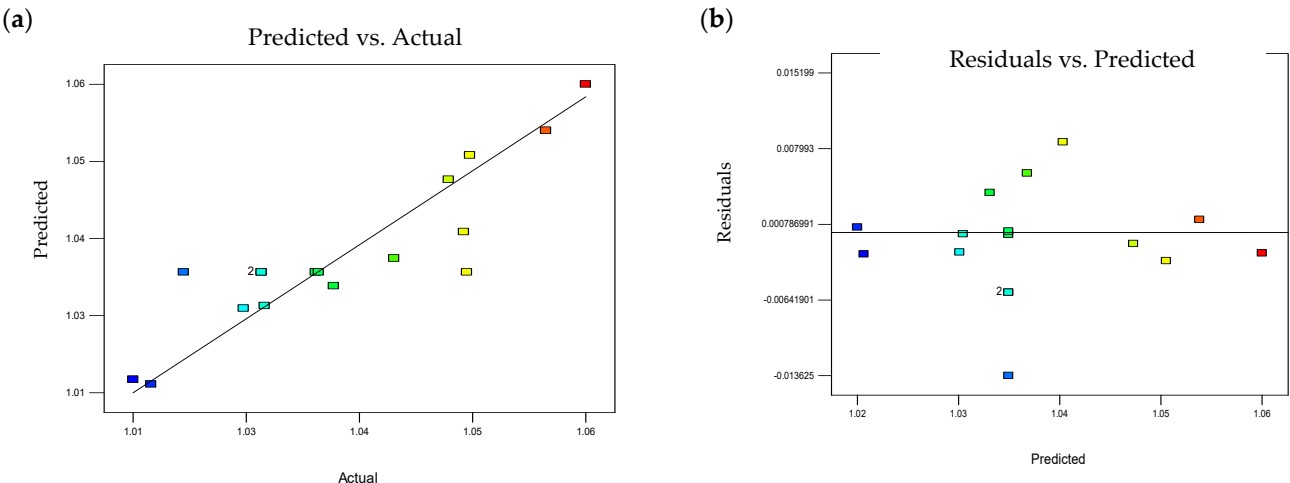

**Figure 10.** CFRP exit delamination analysis for (**a**) actual and predicted plot and (**b**) predicted and residual plot.

### 3.1.2. Effect of Geometric Parameters on Exit Delamination

For exit delamination of the CFRP ($Y_1$) panel, the perturbation plot shown in Figure 11a was used to determine the sensitivity of each factor. The exit delamination was significantly affected by the chisel edge angle (A). The exit delamination of the CFRP was decreased by increasing the chisel edge angle. With these drill geometries, a lower exit delamination was consequently produced. In this parameter analysis, the primary clearance angle (B) had a significantly greater effect than the point angle (C) on the exit delamination ($Y_1$) of the CFRP. According to the quadratic model that was fitted, a curvilinear profile was observed, as shown in Figure 11b. By maintaining the third parameter i.e., chisel edge angle constant at the middle level (37.5°), the graph indicated delamination with regard to two alternative parameters. When the point angle was set at 130° and the primary clearance angle was set at 8°, exit delamination was decreased.

### 3.2. Burr Height Analysis

Exit burr formation was examined using an Alicona optical microscope and burr formation was relatively uniform across the circles of the holes. The development of burrs was likely due to the accumulation of heat from the CFRP panel, which enabled the extrusion of softened Al7075-T6 at the tool margin area. The twist drill's optimal drill geometry for drilling a stack material in a single shot is one that produces the least amount of burrs because adding a second process to remove the formed burrs would raise the cost of the manufacturing process. For burrs formed at the exit of the drilled hole, the deburring process can account for approximately 30% of the total manufacturing cost and can occupy 40% of the total machining time [47]. According to Sakurai et al. [28], a large point angle ensured maximum lip movement as soon as possible to prevent work

hardening, which caused thinner burrs because of a shift in chip flow direction. Low feed rates (0.05 mm/rev) are required to guarantee the least amount of thrust force in order to reduce burr development [48].

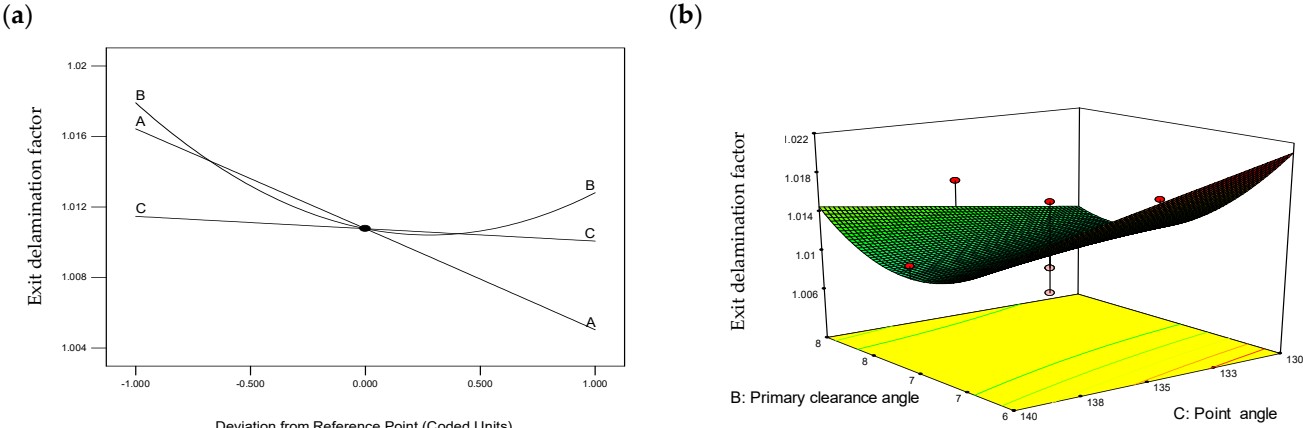

**Figure 11.** (**a**) Perturbation plot and (**b**) 3D Response surface for exit delamination of CFRP. A, chisel edge angle; B, primary clearance angle; C, point angle.

Figure 12 displays the measurements of the maximum and minimum exit burr formation for the typical drilled hole in Al7075-T6. The average $H_{bmax}$ value was minimal, measuring between 40.2 and 271.2 μm, as shown in Figure 13. A lower $H_{bmax}$ value was found in R16 with a 45° chisel edge angle, 6° primary clearance angle, and 130° point angle, whereas R3 yielded a higher $H_{bmax}$ value with a 30° chisel edge angle, 7° primary clearance angle, and 135° point angle [35]. These results are in line with ref. [49] in which the burr height ranged from 133.62 to 211.45 μm when they used a 130° point angle and from 1036.25 to 2066.85 μm when they used a tool with a 110° point angle. Further, these authors mentioned that the drill with a 130° point angle produced a uniform burr type and a 110° point angle produced transient and crown burrs during single-shot drilling of CFRP/Al7075-T6 material [49].

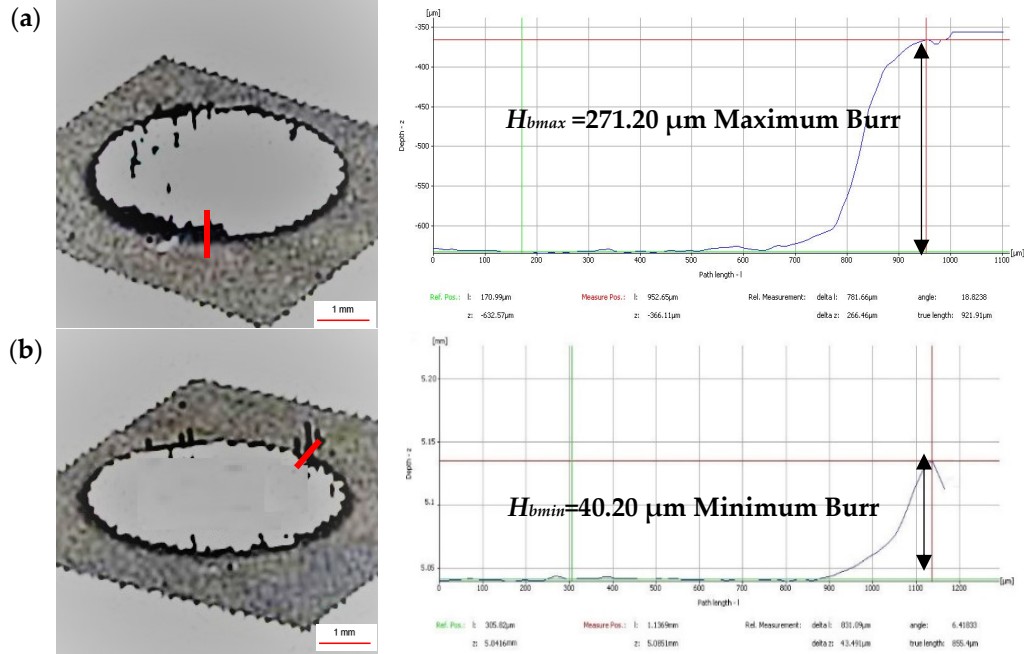

**Figure 12.** (**a**) Maximum and (**b**) minimum burr formation measurements.

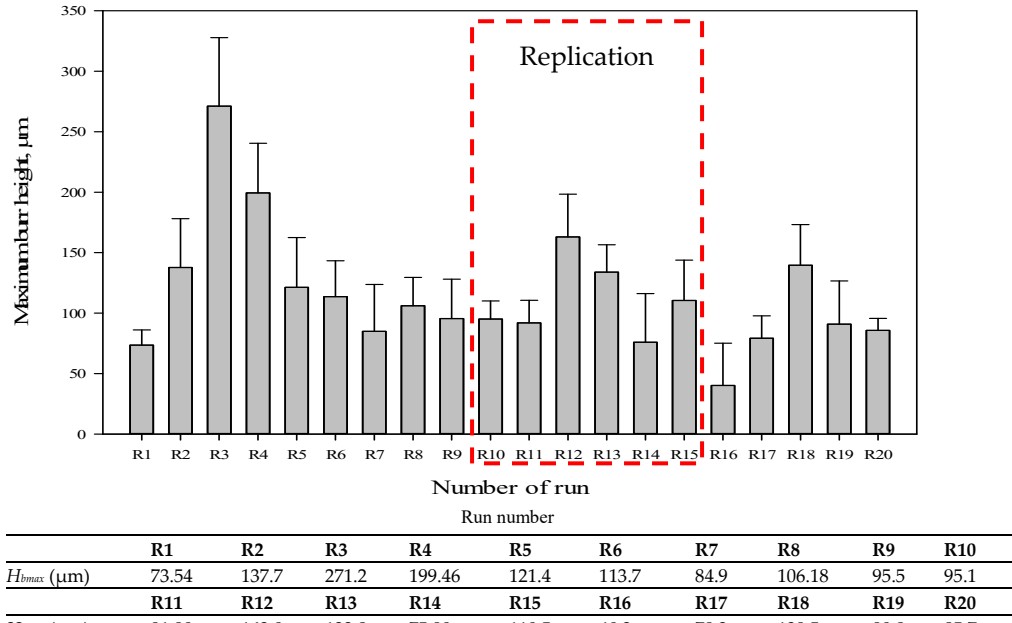

| Run number | | | | | | | | | |
|---|---|---|---|---|---|---|---|---|---|
| | **R1** | **R2** | **R3** | **R4** | **R5** | **R6** | **R7** | **R8** | **R9** | **R10** |
| $H_{bmax}$ (µm) | 73.54 | 137.7 | 271.2 | 199.46 | 121.4 | 113.7 | 84.9 | 106.18 | 95.5 | 95.1 |
| | **R11** | **R12** | **R13** | **R14** | **R15** | **R16** | **R17** | **R18** | **R19** | **R20** |
| $H_{bmax}$ (µm) | 91.90 | 163.0 | 133.8 | 75.90 | 110.5 | 40.2 | 79.2 | 139.5 | 90.8 | 85.7 |

**Figure 13.** Burr formation data during all 20 runs.

When the chisel edge angle was reduced to 30°, the $H_{bmax}$ value rose, resulting in a significant rolled-up phenomenon at the Al707-T6 panel's exit. This is because there was less space for the chip to flow during evacuation when the bit first contacted the material during the cutting operation, because the chisel edge angle of 30° was less than 45°, as shown in Figure 14. The ineffective chip flow increased the shear and decreased the cutting efficiency during the drilling process. The cutting heat also makes the material more ductile and uses more energy [32]. As a result, burrs along the hole's edge are easily produced. The replicated tools yielded consistent results, as shown in Figure 13, proving that they were properly manufactured.

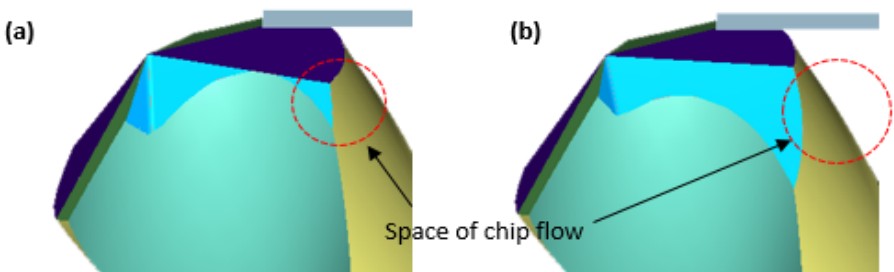

**Figure 14.** Space for chip flow showing two geometries with same primary clearance angle and point angle where (**a**) drill with 30° of chisel edge angle (**b**) drill with 45° of chisel edge angle.

### 3.2.1. Regression Model and ANOVA

To obtain the lowest residuals between the anticipated and actual values for $H_{bmax}$, the regression model for the response was enhanced by log transformation. The final empirical model for the actual causes of burr formation at the Al7075-T6 panel's exit ($Y_2$) was

$$Y_2 = -25.99962 - 0.22321A + 5.9706B + 0.1564C - 0.02019BC + 2.747e^{-3}A^2 - 0.2298B^2 \tag{7}$$

The *F*-value for $H_{bmax}$ in the ANOVA analysis was 9.718 and the *p*-value was lower than 0.05, as shown in Table 5. Furthermore, the *p*-value of 0.683 indicated that the lack of fit was related to pure error and was not significant. The model's significant value and the lack of fit's non-significant value supported the validity of the log-transformed

model. As indicated in Table 5, the factors that significantly influenced the results had a confidence level above 95% and a *p*-value lower than 0.05. The second term, B, was insignificant, despite the fact that the *p*-values for the other model terms (A, C, BC, $A^2$, and $B^2$) had a considerable impact on $H_{bmax}$. The adjusted $R^2$ value of 0.733 and the predicted $R^2$ value of 0.602 were in reasonable accord as the discrepancy was less than 0.2. Since a ratio greater than 4 is preferred when measuring the signal-to-noise ratio [45], as mentioned in Section 3.1.1, and a strong signal of 12.85 was obtained here, this model was utilized to navigate the design space. The correlation coefficient ($R^2$) and adjusted coefficient (Adj $R^2$) values in this instance were 0.818 and 0.733, respectively, demonstrating the significance of the fit of the RSM model and its potential for response prediction.

**Table 5.** Pooled ANOVA of model for maximum burr formation at the exit of Al7075-T6 panel.

| Source | Sum of Squares | df | Mean Square | F Value | *p*-Value Prob > F | PC (%) | |
|---|---|---|---|---|---|---|---|
| **Model ($Y_2$)** | 0.55648 | 6 | 0.08024 | 9.71841 | 0.0004 | | significant |
| Chisel edge angle (A) | 0.16492 | 1 | 0.16492 | 19.97565 | 0.0006 | 24.8% | |
| Primary clearance angle (B) | 0.0077 | 1 | 0.0077 | 0.93271 | 0.3518 | 1.2% | |
| Point angle (C) | 0.0569 | 1 | 0.0569 | 6.89196 | 0.021 | 8.6% | |
| BC | 0.08154 | 1 | 0.08154 | 9.87583 | 0.0078 | 12.3% | |
| $A^2$ | 0.07645 | 1 | 0.07645 | 9.25995 | 0.0094 | 11.5% | |
| $B^2$ | 0.16897 | 1 | 0.16897 | 20.46549 | 0.0006 | 25.5% | |
| Residual | 0.10733 | 13 | 0.00826 | | | 16.2% | |
| Lack of Fit | 0.05707 | 8 | 0.00713 | 0.70959 | 0.6832 | | not significant |
| Pure Error | 0.05026 | 5 | 0.01005 | | | | |
| Cor Total | 0.66381 | 19 | | | | | |
| Std. Dev. | 0.09086 | | $R^2$ | | 0.817699 | | |
| Mean | 2.02779 | | Adj $R^2$ | | 0.733559 | | |
| C.V. % | 4.48093 | | Pred $R^2$ | | 0.601704 | | |
| PRESS | 0.2345 | | Adeq Precision | | 12.8502 | | |

When the actual value gained through experimentation was compared with the predictions of the model, as shown in Figure 15a, it can be observed that the points were evenly split by a 45-degree line, which proved the model fit. Figure 15b demonstrates that, when compared to the expected value predicted by the empirical model, the response of the experimental data was mostly contained within the range of allowable deviations. When drilling stack-up material, the Al7075-T6 burr formation could be estimated using the regression model that was created here. The standard error estimation (SEE) result for $H_{bmax}$ $\log_{10}$ was 0.0732, according to Figure 15b. For example, the actual number fell between 1.9068 and 2.0532, and the anticipated value was 1.98. For a dataset with a normal linear relationship, the RSM model can be used to estimate the value if two-thirds of the residual data points (Figure 15b) are within SEE i.e., above or below the least squares line [50].

### 3.2.2. Effect of Geometric Parameters on Burr Height Formation

For $H_{bmax}$ of the Al7075-T6 panel ($Y_2$), the perturbation plot shown in Figure 16a was used to determine the sensitivity of each factor. The variables had a significant impact on the specific responses in the drilling of stack-up material. The $H_{bmax}$ value was significantly affected by the chisel edge angle (A). The $H_{bmax}$ value at the exit of Al7075-T6 was decreased by increasing the chisel edge angle. An extreme chisel edge angle made clearance possible and made the shearing of materials by the cutting edges more effective (Figure 14). With these drill geometries, less burr formation was consequently produced. In this parameter analysis, the primary clearance angle (B) and point angle (C) had a moderate impact on the response of ($Y_2$).

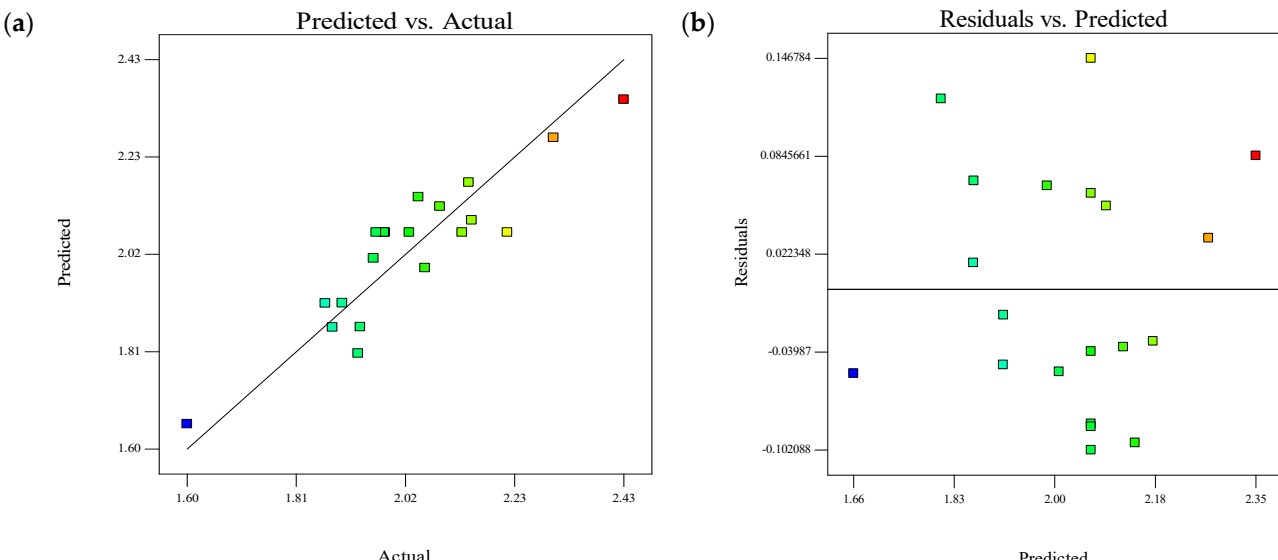

**Figure 15.** Analysis of maximum burr formation on Al7075−T6 for (**a**) actual and predicted plot and (**b**) predicted and residual plot.

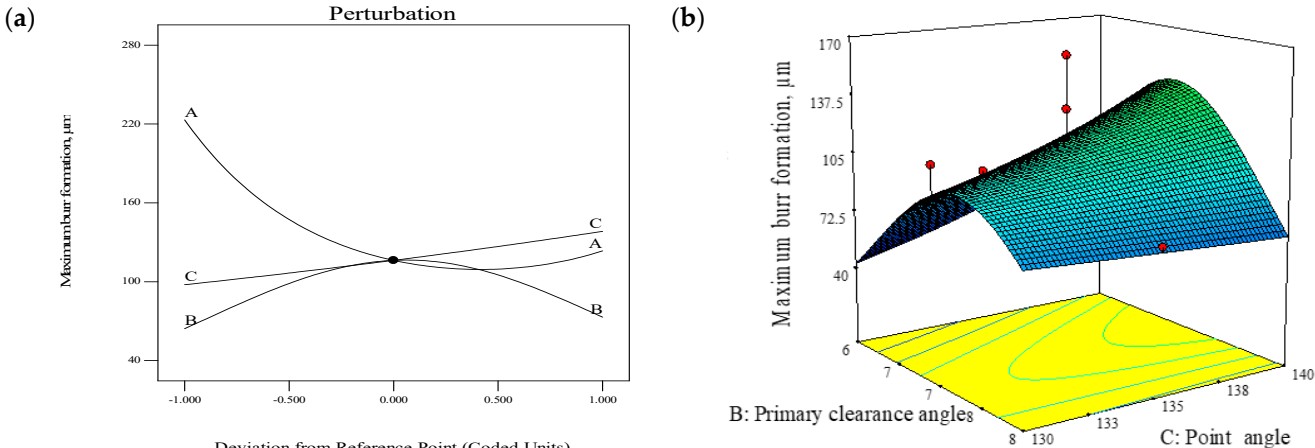

**Figure 16.** (**a**) Perturbation plot and (**b**) 3D response surface for exit burr height at Al7075−T6. A, chisel edge angle; B, primary clearance angle; C, point angle.

Figure 16b displays the 3D surface graphs for burr development of Al7075-T6. According to the quadratic model that was fitted, the results had a curvilinear profile. By maintaining the third parameter i.e., chisel edge angle constant at the middle level (37.5°), the graph indicated the $H_{bmax}$ with regard to two alternative parameters. When minimum point angle and primary clearance angle were set, i.e., primary clearance angle of 6° and point angle of 130°, the $H_{bmax}$ was decreased.

### 3.3. Multiple Response Optimization

This section presents the target response optimization based on the developed regression function of each response connected with the cutter geometry. Optimizing the target response facilitates achieving a set of ideal target response conditions. It can maintain all the desired response ranges or at least optimize all the desired responses. The target response technique seeks to improve quality, cost, and time, while increasing product efficiency. In this study, the target response was optimized using two techniques: an overlay plot and the desirability function. The objectives of this optimization procedure were to create a cutter with the least amount of thrust force and burr development. The goal and limitations for the variables to simultaneously attain many desired goals are tabulated in Table 6.

**Table 6.** Goals and constraints for the factors and responses.

| Contraints | | | |
|---|---|---|---|
| Factor/Response | Goal | Lower Limit | Upper Limit |
| Chisel edge angle (A) | Within range | 30° | 45° |
| Primary clearance angle (B) | Within range | 6° | 8° |
| Point angle (C) | Within range | 130° | 140° |
| Burr Height ($H_{bmax}$) | Minimize | 40.2 μm | 271.2 μm |
| Delamination ($F_{d\text{-}exit}$) | Minimize | 1.0046 | 1.0196 |

Based on the goals, one solution was proposed, as tabulated in Figure 17. A desirability level closer to 1 indicates that the goals are not easy to reach. In other words, a higher desirability index represents the closest response to the target or ideal values. As shown in Figure 17, the proposed solution gave a desirability index of 0.773.

Solutions

| Number | Chisel edge ar | Primary cleara | Point angle | Burr Height | Delamination | Desirability | |
|---|---|---|---|---|---|---|---|
| 1 | 45.0 | 8 | 130 | 82.2307 | 1.00528 | 0.773 | Selected |
| 2 | 44.9 | 8 | 130 | 81.9794 | 1.00532 | 0.773 | |
| 3 | 44.8 | 8 | 130 | 81.4022 | 1.00542 | 0.772 | |
| 4 | 45.0 | 8 | 130 | 82.0924 | 1.00533 | 0.772 | |
| 5 | 44.6 | 8 | 130 | 80.5027 | 1.00559 | 0.771 | |
| 6 | 44.5 | 8 | 130 | 80.245 | 1.00564 | 0.771 | |

**Figure 17.** Proposed solution report for the optimization tool geometry process.

In this this experiment, the optimal cutter geometry (45° chisel edge angle, 8° primary clearance angle, and 130° point angle) was proposed based on a combination of the minimum exit delamination and least amount of burr height, predicted from Equations (6) and (7) to be 1.00528 and 82.2307 μm, respectively. The predicted optimized results for the responses $Y_1$ and $Y_2$ are tabulated in Table 7. For the suggested optimal drill bit shape, the discrepancies between the predicted and actual trial results were 0.11% and 9.72% respectively, hence validating that the proposed optimized cutter geometry was confirmed in the optimization model.

**Table 7.** Prediction of the optimized model of twist drill bit for edge defect analysis when drilling CFRP/Al7075-T6 stack-up material.

| Responses (Y) | $Y_1$, [μm] | $Y_2$, [μm] |
|---|---|---|
| Model response | 1.00528 | 82.2307 |
| Experimental | 1.00635 | 74.234 |
| Error (%) | 0.11 | 9.72 |

*3.4. Hole Diameter Error*

The hole accuracy attained here was in accordance with industry standards. This means the diameter tolerances fell within the range of the H8 zone, i.e., 18 μm [31,51]. The evaluation of the hole diameter error is shown in Figure 18 for both CFRP and Al7075-T6 plates. The absolute difference between the measured value and nominal value is the diameter error for CFRP and Al7075-T6. The variations in panel diameters between CFRP and Al7075-T6 are also noted and given the name "stack up diameter error".

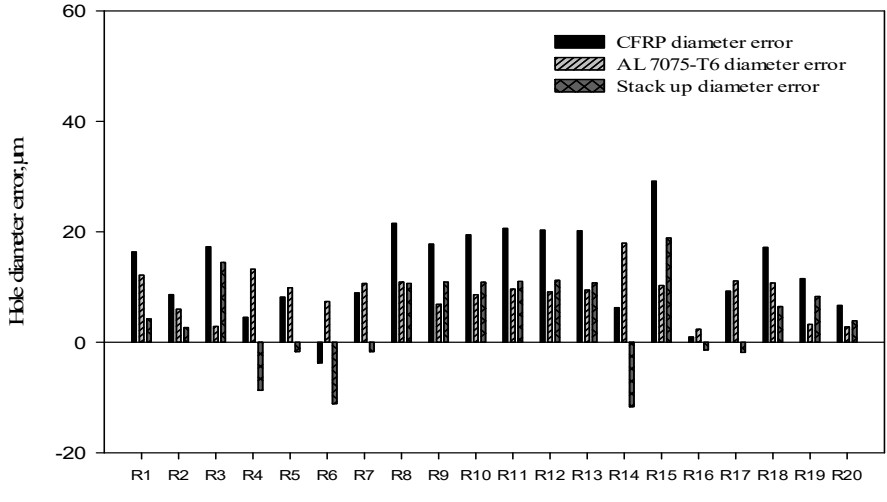

| | **R1** | **R2** | **R3** | **R4** | **R5** | **R6** | **R7** | **R8** | **R9** | **R10** |
|---|---|---|---|---|---|---|---|---|---|---|
| $e_{a\ CFRP}$, [μm] | 16.38 | 8.64 | 17.32 | 4.50 | 8.18 | −3.76 | 8.96 | 21.56 | 17.78 | 19.48 |
| $e_{a\ Al}$, [μm] | 12.14 | 5.98 | 2.86 | 13.22 | 9.88 | 7.38 | 10.66 | 10.90 | 6.88 | 8.60 |
| $e_{a\ Stack}$, [μm] | 4.24 | 2.66 | 14.46 | −8.72 | −1.70 | −11.14 | −1.70 | 10.66 | 10.90 | 10.88 |
| | **R11** | **R12** | **R13** | **R14** | **R15** | **R16** | **R17** | **R18** | **R19** | **R20** |
| $E_{a\ CFRP}$, [μm] | 20.64 | 20.34 | 20.22 | 6.28 | 29.20 | 0.96 | 9.26 | 17.20 | 11.54 | 6.68 |
| $e_{a\ Al}$, [μm] | 9.62 | 9.12 | 9.46 | 17.96 | 10.28 | 2.36 | 11.10 | 10.72 | 3.23 | 2.78 |
| $e_{a\ Stack}$, [μm] | 11.02 | 11.22 | 10.76 | −11.68 | 18.92 | −1.40 | −1.84 | 6.48 | 8.31 | 3.90 |

**Figure 18.** Hole diameter errors between stack-up materials for all runs.

According to the graph, the hole diameter of the CFRP material was found to be undersized only in R6 with a value of −3.76 μm. This is because a shrinking effect was induced by the drill geometry during the cutting operation [ref]. The cutting performance when drilling the CFRP panel would be decreased due to the narrower primary clearance angle. To limit the amount of shrinkage when drilling composite panels, a high clearance is required. The range for oversized holes in this research was 0.96 to 29.2 μm. Soo et al. [31] obtained a similar range of oversized holes between 6 to 34 μm while drilling CFRP/AA7010-T7451 with 6.38 mm flat point drills. Overall, R8, R10, R11, R12, R13, and R15 [35] did not meet the customer's specifications since one of the errors was greater than the OEM standard's permitted maximum. R16, the ideal cutter geometry, showed the least error for CFRP, Al7075-T6, and stack-up material. In all of the runs that were analyzed for stack-up error, values ranged from −1.40 to 18.92 μm.

When comparing individual hole diameters for CFRP and Al7075-T6, in some cases the hole diameters of Al7075-T6 were larger than those of CFRP and in some cases it was other way round. Both of these results were obtained in the past by various researchers, with different explanations. Soo et al. [31] mentioned that while drilling CFRP/AA7010-T7451 aluminum with a flat point drill, the Al layer hole was larger than the CFRP layer hole, as the former has a lower modulus of elasticity and higher thermal expansion compared to the latter [20]. Distinct elastic modulus values experience varying levels of elastic deformation during drilling, resulting in different hole dimensions. Additionally, when drilling Al7075-T6, the aluminum chips clogged the flute due to the specific geometry and raised the drilling temperature [52]. A smaller CFRP hole diameter is also due to the fact that the fibers flex back into the hole after a few days [53]. The cases where the hole diameter of CFRP was larger than that of Al7075-T6 are largely associated with continuous chip formation. Continuous chips likely twist along the drill body, leading to clogging. Thus, hot, sharp chips that are unable to be smoothly evacuated remain in the hole, enlarging the CFRP holes and deteriorating the surface quality of the CFRP.

### 3.5. Hole Circularity

Figure 19 displays the average hole circularity when drilling CFRP/Al7075-T6 stack material using various drill geometries. Overall, the average hole circularity of the CFRP laminates was better, ranging from 13.40 to 24.97 μm, compared to 13.23 to 26.07 μm for Al7075-T6. The results obtained here were better than the results obtained by ref. [31] in which drilling CFRP/AA7010-T7451 aluminum alloy gave values varying from 5 to 45 μm. In addition, the hole circularity error values obtained in the CFRP layer were generally larger than those in the Al7075 section [31], which was possibly due to tool runout causing radial deflection or initial chisel edge sliding (also known as 'walking') as the drill penetrated the top CFRP layer. No 'walking' was found in the current experiment, as no such trend in hole circularity errors was observed between CFRP and Al7075-T6.

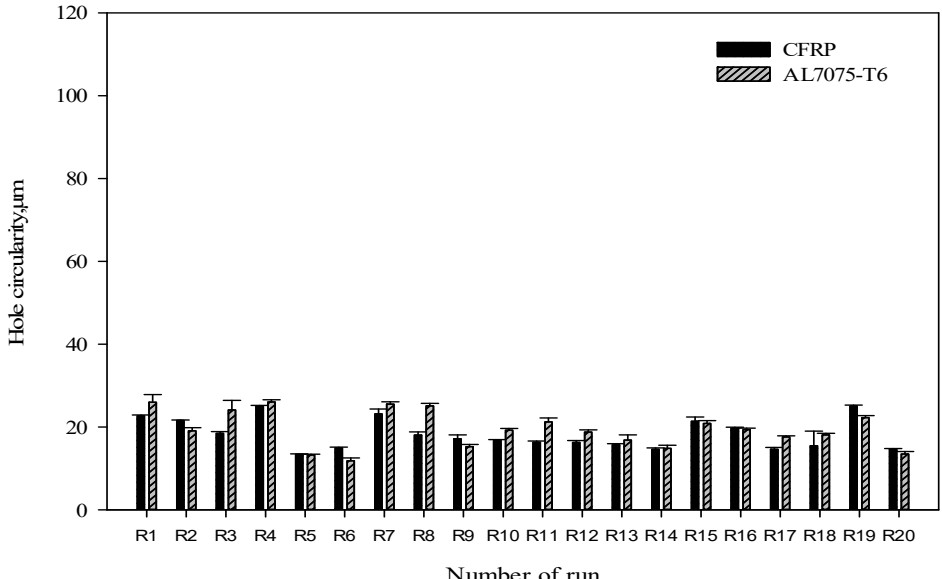

|  | R1 | R2 | R3 | R4 | R5 | R6 | R7 | R8 | R9 | R10 |
|---|---|---|---|---|---|---|---|---|---|---|
| Circularity$_{CFRP}$, [μm] | 22.67 | 21.60 | 18.47 | 24.97 | 13.40 | 14.97 | 23.20 | 18.07 | 17.17 | 16.73 |
| Circularity$_{Al}$, [μm] | 25.97 | 19.10 | 24.13 | 26.07 | 13.23 | 11.83 | 25.53 | 25.10 | 15.27 | 19.23 |
|  | R11 | R12 | R13 | R14 | R15 | R16 | R17 | R18 | R19 | R20 |
| Circularity$_{CFRP}$, [μm] | 16.30 | 16.27 | 15.70 | 14.60 | 21.40 | 19.70 | 14.60 | 15.50 | 25.03 | 14.57 |
| Circularity$_{Al}$, [μm] | 21.27 | 18.77 | 16.83 | 14.87 | 20.83 | 19.37 | 17.60 | 18.10 | 22.27 | 13.47 |

**Figure 19.** Hole circularity between stack-up materials for all runs in the extended study.

For all runs, the influence of the selected drilling parameter (2600 rev/min and 0.05 mm/rev) produced good results according to OEM standards. The smallest hole circularity error was found in R5 for CFRP (primary clearance angle = 8°, point angle = 140°, and chisel edge angle = 30°) and R6 for Al7075-T6 (primary clearance angle = 7°, point angle = 130°, and chisel edge angle = 37.5°). When the Al7075-T6 first encountered the drill bit when drilling the stack material, the stability was increased by reducing the tip angle to 130°. As a result, when the cutting tool needed to cut through the stack of material, there were less deflections and vibrations.

## 4. Conclusions

Detailed research, comprising experimentation, analysis, regression model construction, and optimization of the unique WC twist drill geometry, was successfully conducted to address hole edge defects and hole integrity of CFRP/Al7075-T6 stack-up material. The drilled hole in a CFRP panel can simply develop delamination while the drilled hole in an aluminum panel can be oversized without careful tool geometry and drilling parameter selection. As a result, the panel will be scrapped and the subsequent assembly procedure

must be discontinued. Therefore, the ideal way to enhance the drilling process is to optimize tool geometry in order to boost drilling productivity and decrease the rejection of drilled parts.

- Although the point angle of the twist drill had to be raised from 130° to 140°, the delamination at the CFRP exit hole had a favorable effect on hole integrity.
- The average burr height was minimal, measuring between 40.2 and 271.2 μm. The lower burr height was found with a 45° chisel edge angle, 6° primary clearance angle, and 130° point angle. When the chisel edge angle was reduced to 30°, the burr height rose, resulting in a significant rolled-up phenomenon at the Al707-T6 panel's exit hole due to less available space for chip evacuation.
- The lowest hole diameter error values were obtained with values of 0.96 μm, 2.36 mm, and -1.4 μm for the stack-up diameter error, CFRP diameter error, and Al7075-T6 diameter error, respectively. At the same time, the hole circularity error was less than 30 μm in all runs, which was within OEM standards.
- Multiple response optimization was employed to optimize drill geometric parameters and the best drill geometry for a customized twist drill was proposed. To obtain minimal hole edge defects, it was discovered that the combination of 45° chisel edge angle, 8° primary clearance angle, and 130° point angle is the ideal drill geometry for a twist drill design, with a desirability index level of 0.773.

**Author Contributions:** Conceptualization, M.H.H. and J.J.M.; methodology, M.H.H.; validation, M.H.H. and J.X.; formal analysis, G.F.; investigation, G.F.; writing—original draft preparation, M.H.H. and J.J.M.; writing—review and editing, J.X., M.H.H. and G.F.; visualization, G.F.; supervision, M.H.H.; project administration, J.X. and M.H.H.; funding acquisition, J.X. and M.H.H. All authors have read and agreed to the published version of the manuscript.

**Funding:** This research was funded by a Short Term Grant (#304/PMEKANIK/6315557). The work was also partly funded by the National Natural Science Foundation of China (Grant Nos. 51705319 and 52175425).

**Data Availability Statement:** Not applicable.

**Acknowledgments:** The authors would like to acknowledge the support from Short Term Grant (#304/PMEKANIK/6315557) for financing this research work. They are also grateful for the financial participation of National Natural Science Foundation of China. Technical support from the School of Mechanical Engineering at USM is also greatly acknowledged.

**Conflicts of Interest:** The authors declare no conflict of interest.

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
