# Peer review of "Hole Quality Observation in Single-Shot Drilling of CFRP/Al7075-T6 Composite Metal Stacks Using Customized Twist Drill Design"

_jcs, doi:10.3390/jcs6120378_

Round 1

Reviewer 1 Report

This research fabricated tungsten carbide (WC) twist drills with various geometric features, and studied the single effect of geometry parameters of drill bit on the hole quality. This work provides the relation between tool geometry and hole quality in single-shot drilling of composite-metal stacks, and it is useful for the single-shot drilling of CFRP/Al7075-T6 stack for the manufacture of aircraft. The paper is written well and the data is sufficient.

While the details need to be addressed such as lines 159, 224, 271, 273, 378 “Error! Reference source not found”. The figure label in line 406 “Figure 1” is wrong. Besides, the resolution of figures in the whole paper needs to be improved.

Author Response

Dear Reviewers,

Thanks for the valuable comments. We already answer all the comments accordingly. The correction is highlighted with the yellow color in the manuscript.

Reviewer 1

While the details need to be addressed such as lines 159, 224, 271, 273, 378 “Error! Reference source not found”. The figure label in line 406 “Figure 1” is wrong. Besides, the resolution of figures in the paper needs to be improved.

Author response

“Error! Reference source not found” is removed from the entire manuscript. Line 406 shows “Figure 11” which is depicted in Line 418. The resolution of the images in the manuscripts is improved from 220ppi to high fidelity.

Reviewer 2 Report

The manuscript discusses the hole quality in terms of hole edge defects and hole integrity with respect to tool geometry. The findings suggested that, within the range of parameters examined, the proposed correlation models might be utilized to predict performance measures. But I would like to comment some particular points appearing in the text:

Point 1: (line 53) Review English construction of the sentence"assure secure assembly with other components for improved product integrity, reliability, and life cycle”.

Point 2: (line 124-125) Please check the grammar of these sentences.

Point 3: Please add a scale to Figures 2 ,4, and 8.

Point 4: I think equation (2) is not really useful. Maybe it can be removed.

Point 5: Color height scale cannot be seen in Figure 3(b).

Point 6: (line 298-299) Please check these sentences.

Point 7: Please check the full text. The sentence of “Error! Reference source not found” appears too many times.

Point 8: Detailed experiments, measurement and optimization are introduced in this manuscript. Please give more explanation and discussion about the effect of geometry parameters on the delamination and burr height. 

Author Response

Dear Reviewers,

Thanks for the valuable comments and we already answer all the comments accordingly. The correction is highlighted with the yellow color in the manuscript.

Reviewer 2

The manuscript discusses the hole quality in terms of hole edge defects and hole integrity with respect to tool geometry. The findings suggested that, within the range of parameters examined, the proposed correlation models might be utilized to predict performance measures. But I would like to comment some particular points appearing in the text:

Point 1: (line 53) Review English construction of the sentence "assure secure assembly with other components for improved product integrity, reliability, and life cycle”.

Author response – Sentence restructured.

Point 2: (lines 124-125) Please check the grammar of these sentences.

Author response – Grammar corrected.

Point 3: Please add a scale to Figures 2 ,4, and 8.

Author response – Scale added to Figures 2 ,4, and 8.

Point 4: I think equation (2) is not really useful. Maybe it can be removed.

Author response – Equation – 2 removed

Point 5: The color height scale cannot be seen in Figure 3(b).

Author response- The author already replace Figure 3(b) with a higher-resolution image

Point 6: (lines 298-299) Please check these sentences.

Author response – Sentence restructured.

Point 7: Please check the full text. The sentence of “Error! Reference source not found” appears too many times.

Author response – “Error! Reference source not found” is removed from the entire manuscript.

Point 8: Detailed experiments, measurement, and optimization are introduced in this manuscript. Please give more explanation and discussion about the effect of geometry parameters on the delamination and burr height. 

Author response- The author already explains the effect of geometry via statistical study. The reviewer may refer to lines 409- 420 for exit delamination and line 519 – 533 for maximum burr height.

Round 2

Reviewer 1 Report

The questions have been carefully revised, and the revised manuscript looks well.